# Pyrotinib versus lapatinib therapy for HER2 positive metastatic breast cancer patients after first-line treatment failure: A meta-analysis and systematic review

**Ye Yuan** *, **Xumei Liu, Yi Cai, Wenyuan Li**

Hospital of Chengdu University of Traditional Chinese Medicine, Chengdu, China

* 981735645@qq.com

**Data Availability Statement:** All relevant data are within the manuscript and its Supporting Information files.

## Abstract

### Introduction

It is critical to select subsequent treatments for patients after the failure of trastuzumab therapy. Following the failure of standard trastuzumab therapy guidelines in the Chinese Society of Clinical Oncology, pyrotinib and capecitabine is a grade I recommended regimen for treating patients with HER2-positive metastatic breast cancer. Concurrently, in treating patients with HER2-positive metastatic breast cancer, lapatinib and capecitabine are also recommended regimens for those who have previously received taxanes, anthracyclines, and trastuzumab therapy. However, there is currently no systematic review and meta-analysis comparing pyrotinib with lapatinib among HER2+ MBC patients. Therefore, this study aims to perform a systematic review and meta-analysis and assess whether pyrotinib is superior to lapatinib in efficacy and safety.

### Methods

Relevant trials were searched in CNKI, Wanfang, VIP, PubMed, Embase, and Cochrane CENTRAL databases from inception until March 27th, 2022. The primary outcomes were PFS and OS, and the secondary outcomes were ORR and grade ≥3 AEs.

### Results

Five relevant studies were included in this study, including 2 RCTs and 3 retrospective cohort studies. Pyrotinib combined with chemotherapy is superior to lapatinib combined with chemotherapy among HER2+ metastatic breast cancer patients, with a significant improvement in PFS (prior trastuzumab therapy) (HR: 0.47, 95% CI: 0.39–0.57, $p<0.001$, $I^2 = 0\%$, FEM), PFS (trastuzumab resistance) (HR: 0.52, 95% CI: 0.39–0.68, $p<0.001$, $I^2 = 40\%$, FEM) and ORR (RR: 1.45, 95% CI: 1.26–1.67, $p<0.001$, $I^2 = 8\%$, FEM), but has higher grade ≥3 diarrhea incidence (RR: 2.68, 95% CI: 1.85–3.90, $p<0.001$, $I^2 = 44\%$, FEM).

**Funding:** The author(s) received no specific funding for this work.

**Competing interests:** The authors have declared that no competing interests exist.

## Conclusions

The efficacy of pyrotinib combined with chemotherapy is superior to lapatinib combined with chemotherapy but has more safety risks.

## Introduction

Globally, breast cancer is the major cause of morbidity and mortality among women, responsible for one-quarter of all new cancer cases and 15% of all cancer deaths, and ninety percent of breast cancer deaths result from metastases at distant sites [1,2]. In addition, breast cancer incidence has increased globally over the last several decades, with the most pronounced increases occurring in previously low-incidence areas [3]. Breast cancer disease is heterogeneous, with different molecular subtypes, including HER2-positive breast cancer, which is defined as a type that has amplified or overexpressed the HER2 gene (or ErbB2) [4]. About 15%-20% of all breast cancers are classified as HER2-positive (+) breast cancers, which is considered an aggressive subtype [5]. In the past, HER2-positive breast cancer has been associated with a higher stage at presentation, higher relapse rates, and a greater risk of breast cancer mortality when not treated with specific HER2 strategies [6]. However, developing HER2-targeted therapies has considerably improved outcomes over the past two decades, including trastuzumab, pertuzumab, lapatinib, and ado-trastuzumab emtansine [7].

Standard first-line treatment is the monoclonal antibodies pertuzumab and trastuzumab combined with a taxane in patients with metastatic disease [8]. However, resistance to trastuzumab develops when either the active target receptor or a component downstream of PI3K/Akt/mTOR pathway is altered [9]. Therefore, additional anti-HER2 drugs are urgently required to treat patients who have previously received treatment with trastuzumab or pertuzumab. Lapatinib and capecitabine are also recommended regimens for patients with HER2-positive metastatic breast cancer who have previously received taxanes, anthracyclines, and trastuzumab therapy. [10]. Lapatinib, a tyrosine kinase inhibitor (TKI), plays an anti-tumor role by competing with intracellular ATP to block HER2 signal, thus blocking phosphorylation and downstream changes in molecular pathways [11]. In WJOG6110B/ELTOP, the lapatinib plus capecitabine arm showed longer progression-free survival (PFS) and overall survival (OS) than trastuzumab plus capecitabine arm among metastatic breast cancer (MBC) patients those who had previous taxane treatment and progressed to trastuzumab-containing regimens [12]. In CEREBEL, the lapatinib plus capecitabine arm showed longer median PFS than the trastuzumab plus capecitabine arm among MBC patients previously treated with trastuzumab [13]. Meanwhile, lapatinib was well tolerated, with both arms showing a similar incidence of grade 3 or grade 4 adverse events [12,13].

Pyrotinib, a second-generation tyrosine kinase inhibitor, inhibits pan-ErbB receptor tyrosine kinases by orally targeting HER1, HER2, and HER4 [14]. The combination of pyrotinib and capecitabine is one of the grade-I recommended regimens for HER2-positive metastatic breast cancer after the failure of standard trastuzumab therapy guidelines of the Chinese Society of Clinical Oncology (CSCO) [15]. Fei Ma et al. reported that a higher overall response rate was found in the pyrotinib arm compared with the lapatinib arm, and the pyrotinib arm showed longer median PFS than lapatinib arm when treated prior trastuzumab therapy [16]. In PHOEBE, patients with pyrotinib showed significantly longer PFS and median PFS than patients with lapatinib [17].

There is currently no relevant systematic review meta-analysis and meta-analysis about the treatment of pyrotinib among MBC patients treated with trastuzumab. However, comparing

the pyrotinib with lapatinib is clinically significant, which may help clinicians and patients select superior treatment. Therefore, this study aims to perform a systematic review and meta-analysis and assess whether pyrotinib is superior to lapatinib in efficacy, especially in long-term survival. Safety will be mentioned, but it is not the main objective of this study.

## Materials and methods

### Study design

The Preferred Reporting Items for Systematic Reviews and Meta-Analyses (PRISMA) guidelines were followed during the conduct of this study [18], and the study was registered in PROSPERO (CRD42022323376). The protocol was not prepared.

### Search strategy

Researchers YY and LXM searched electronic databases including PubMed, Embase, the Cochrane Library, CNKI, Wan Fang, and Sinomed from inception until March 27th, 2022 for relevant studies. All data can be found in public repository. The Chinese search terms were "ruxianai," "ruxianzhongliu," "ruai," "bigetini," and "lapatini." The English search terms are shown in Table 1.

### Inclusion criteria

The inclusion criteria in this study were based on i) clinical, histological, or pathological diagnoses, patients with HER2 positive breast cancer (fluorescent in situ hybridization (FISH), immunohistochemistry, or both stained positively 3+); ii) treatment of P or L arms with chemotherapy combined with pyrotinib or lapatinib; iii) primary outcomes: PFS, OS; secondary outcomes: overall response rate (ORR), and grade ≥3 adverse events (AEs); and iv) study design: RCTs, cohort studies, and retrospective studies. v) articles that studied the two medications as second-line after treatment failure.

### Exclusion criteria

The exclusion criteria were i) conference abstracts and letters, among others, and ii) studies with unavailable outcomes.

### Study selection

The retrieved results were exported to NoteExpress 3.2.0, and duplicate records were searched and deleted. Then, the titles and abstracts of all papers were initially screened, and the irrelevant records were removed. Finally, the rest of the studies were downloaded for further screening, and the studies that did not meet inclusion criteria or had no outcomes available were excluded. The screening results of two researchers were checked for consistency. Whenever a

**Table 1. Search strategy of pyrotinib vs. lapatinib therapy for HER2 positive breast cancer.**

| Number | Search terms |
|---|---|
| **#1** | "neoplasm"[Title/Abstract] OR "carcinoma"[Title/Abstract] OR "cancer"[Title/Abstract] OR "tumor"[Title/Abstract] |
| **#2** | "breast"[Title/Abstract] |
| **#3** | "lapatinib"[Title/Abstract] OR "Tykerb"[Title/Abstract] |
| **#4** | "pyrotinib"[Title/Abstract] |
| **#5** | #1 AND #2 AND #3 AND #4 |

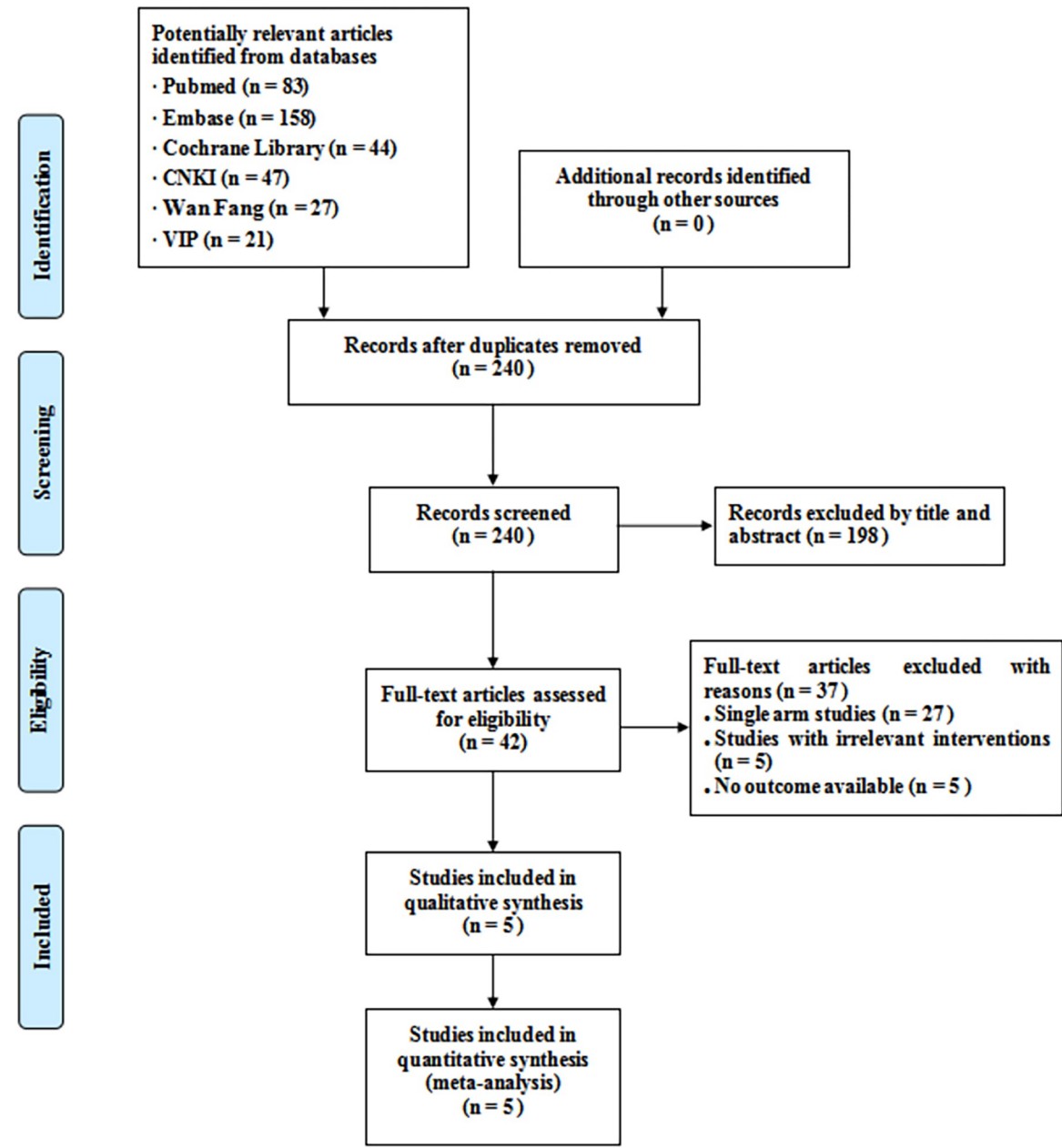

**Fig 1. PRISMA diagram of literature searching and screening process.**

disagreement occurs, a third researcher will be consulted. The process of study selection is shown in Fig 1.

## Data extraction and quality assessment

Two researchers extracted the relevant information using a predefined data extraction table containing (see data extraction table) basic literature information (trial name, title, author, registration number, and publication year), demographic information (number of participants in P arm and L arm, number of trastuzumab resistance participants, tumor stage and diagnosis of patients, inclusion and exclusion criteria) and intervention feature information (duration and

dose of chemotherapy and anti-HER 2 therapy). Randomized control trials (RCTs) will be analyzed using the Cochrane Risk of Bias Tool (https://methods.cochrane.org/bias/resources/cochrane-risk-bias-tool). In addition, study quality was assessed according to Newcastle Ottawa's quality assessment scale for cohort and retrospective studies. The highest score was 9, including 4 points for population selection, 2 points for comparability, and 3 points for population exposure. A score above 6 was considered high quality. Whenever a disagreement occurred, a third researcher was consulted.

### Statistical analysis and evidence quality assessment

All analyses were conducted using RevMan 5.3 and Stata 14. The inverse variance method was used to calculate a pooled hazard ratio (HR) for PFS and OS, and RRs were calculated for dichotomous outcomes, including ORR and grade$\geq$3 AEs. An analysis of heterogeneity was conducted with the $\chi$2 test and $I^2$ statistics. We used a fixed-effect model for analyzing effect quantities with p = 0.1 and $I^2$ = 50%, indicating acceptable heterogeneity. Alternatively, the random-effects model was used when P > 0.1 or $I^2$ > 50% indicated significant heterogeneity.

According to the study design, a random-effects model was used to analyze subgroups (RCTs vs. non RCTs). First, sensitivity analysis was performed with the leave-one-out procedure to uncover the heterogeneity in the primary outcomes. Then, a fixed-effects model was used after removing evident heterogeneity studies to obtain a new effect, which was compared to the previous effect to observe whether a significant difference exists. A publication bias was detected with Egger's test and taken into account when p $\leq$ 0.05. Five aspects (risk bias, imprecision, indirectness, publication bias, and inconsistency) of evidence quality were assessed using GRADE profiler 3.6. We assessed evidence quality as high quality, medium quality, low quality, or extremely low quality.

## Results

### Study selection and characteristics

A total of 380 relevant articles were identified from databases. First, 240 records were identified after removing duplicated records. Then, after removing the records that were not eligible based on titles and abstracts, 198 studies were removed. Then, after reading the full-text articles, 37 studies were excluded. In addition, 27 studies were single-arm studies, 5 studies had irrelevant interventions, and 5 studies had no outcome available. Finally, 5 studies were identified (Fig 1).

The included studies, including 2 RCTs and 3 retrospective studies, were published from 2019 to 2021. A total of 843 participants (P arm: 392, L arm: 451) were involved, including 353 participants with trastuzumab resistance, and the median follow-up time varied from 9.7 months to 27 months. The main characteristics of the selected studies are shown in Table 2.

### Quality assessment of the included studies

Two RCTs (Binghe Xu 2021 and Fei Ma 2019) described the random sequence generation and why participants withdraw and exit. Binghe Xu (2021) made detailed illustrations of allocation concealment, while Fei Ma 2019 made it unclear. The two RCTs did not perform blinding methods (Figs 2 and 3). The studies of Huihui Yang (2021), Yizhao Xie (2021), and Fei Chen (2021) did not receive scores for representativeness of the exposed cohort and assessment of outcome, and Fei Chen (2021) did not have an adequate cohort follow-up. Overall, the studies of Huihui Yang (2021) and Yizhao Xie (2021) received 7 points for high quality, while Fei Chen (2021) received 6 points for low quality (Table 3).

**Table 2. Main characteristics of the selected studies.**

| Author | Study design | MF | | Interventions | | Sample size | | Trastuzumab resistance | | Outcomes |
|---|---|---|---|---|---|---|---|---|---|---|
| | | P arm | L arm | P arm | L arm | P arm | L arm | P arm | L arm | |
| Binghe Xu 2021 [17] | RCT | 10.5 months | 9.7 months | Oral pyrotinib 400mg once daily + oral capecitabine 1000 mg/m² twice daily d1-d14 q21d | Oral lapatinib 1250 mg once daily + oral capecitabine 1000 mg/m² twice daily d1-d14 q21d | 134 | 132 | 37 | 32 | a, c, d |
| Fei Ma 2019 [16] | RCT | 14.9 months | | Oral pyrotinib 400mg once daily + oral capecitabine 1000 mg/m² twice daily d1-d14 q21d | Oral lapatinib 1250 mg once daily + oral capecitabine 1000 mg/m² twice daily d1-d14 q21d | 65 | 63 | NR | | a, b, c, d |
| Huihui Yang 2021 [19] | Retrospective cohort study | NR | | Oral pyrotinib 320mg once daily + chemotherapy | Oral lapatinib 250mg once daily + chemotherapy | 68 | 96 | 68 | 96 | a, c, d |
| Yizhao Xie 2021 [20] | Retrospective cohort study | 20 months | | Pyrotinib (320–400 mg/day) + vinorelbine (25mg/m² intravenously or 60 mg/m² orally on d1 and 8 q21d) | Lapatinib (750–1,250 mg/day) + capecitabine (1,500–2,000 mg/m²) | 92 | 132 | 30 | 29 | a, d |
| Fei Chen 2021 [21] | Retrospective cohort study | 13 months | 27 months | Pyrotinib + capecitabine | Lapatinib + capecitabine | 33 | 28 | 33 | 28 | c, d |

NR: Not reported; MF: Median follow-up; P: Pyrotinib; L: Lapatinib; RCT: Randomized controlled trial; a: PFS; b: OS; c: ORR; d: Grade≥3 AEs.

## Primary outcomes

**PFS.** Four studies [16,17,19,20] reported data on PFS (defined as the time from randomization to first disease progression or death from any cause) for pooling in meta-analysis. First, we pooled all data from participants with prior trastuzumab. Heterogeneity tests of $p = 0.40$ and $I^2 = 0$ were tested in PFS, showing no heterogeneity. Therefore, fixed-effect models were used. The P arm showed significant improvements in PFS compared to the L arm (HR: 0.47, 95% CI: 0.39–0.57, $p<0.001$; Fig 4). In addition, we pooled all data from participants with trastuzumab resistance. Heterogeneity tests of $p = 0.19$ and $I^2 = 40\%$ were tested in PFS, showing low heterogeneity. Therefore, fixed-effect models were used. The P arm showed significant improvements in PFS compared to the L arm (HR: 0.52, 95% CI: 0.39–0.68, $p<0.001$; Fig 4).

*Subgroup analysis.* The P arm showed significant improvements in PFS of RCTs compared to the L arm (HR: 0.39, 95% CI: 0.28–0.53, $p<0.001$). The P arm showed significant improvements in PFS of non RCTs compared to the L arm (HR: 0.52, 95% CI: 0.41–0.66, $p<0.001$). No subgroup difference was found (interaction test, $p = 0.15$).

**OS.** Only Fei Ma 2019 reported the HR of OS (HR: 0.69, 95% CI: 0.40–1.19). In Binghe Xu 2021, the median OS of the P arm was 26.8 months (95% CI 26.2–not reached), and the L arm

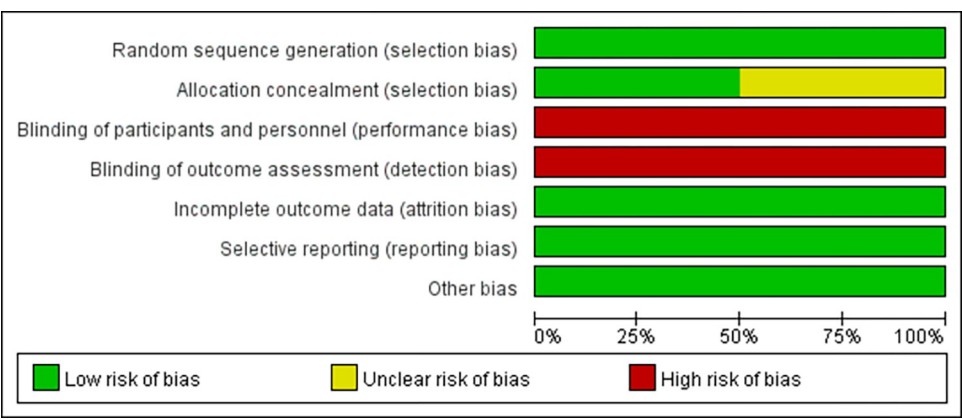

**Fig 2. Risk of bias graph.**

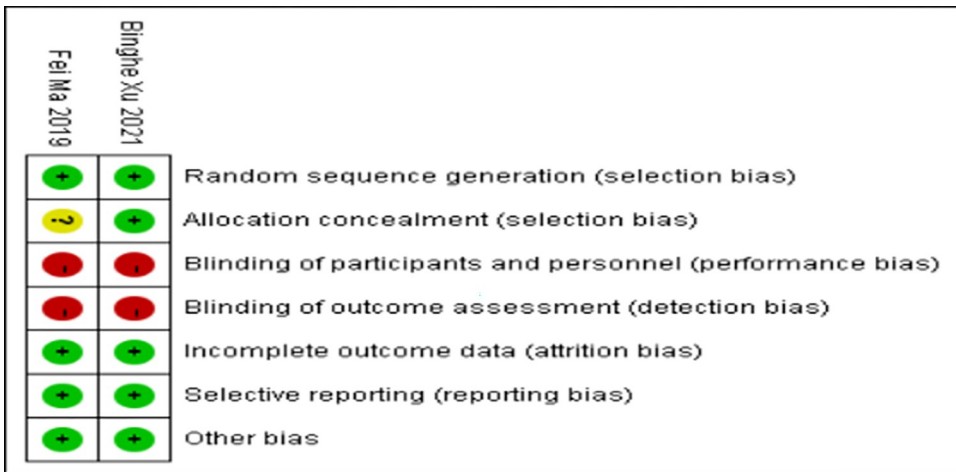

**Fig 3. Risk of bias summary.**

was not reached (21.8–not reached). While, in Fei Ma 2019, the median OS of the P arm was not reached (95% CI 26.3–not reached) and of the L arm was 29.9 months (23.7–not reached).

## Secondary outcomes

**ORR.** Four studies [16,17,19,21] reported data on ORR (the proportion of patients with the best overall response of complete or partial response for at least three months) for pooling in meta-analysis. Heterogeneity tests of p = 0.35 and $I^2$ = 8% were tested in ORR, showing a low heterogeneity. Therefore, fixed-effect models were used. The P arm showed significant improvements in ORR compared to the L arm (RR: 1.45, 95% CI: 1.26–1.67, $p$<0.001; Fig 5).

*Subgroup analysis.* The P arm showed significant improvements in ORR of RCTs compared to the L arm (RR: 1.33, 95% CI: 1.14–1.56, $p$ = 0.0004). The P arm showed significant improvements in ORR of non RCTs compared to the L arm (RR: 1.78, 95% CI: 1.34–2.38, $p$<0.001). No subgroup differences were found (interaction test, $p$ = 0.08).

**Grade≥3 AEs.** Data on grade≥3 adverse events reported in more than half of the trials were obtained. In addition, four trials [16,17,19,20] provided data on adverse events (Adverse events were graded according to the National Cancer Institute Common Terminology Criteria for Adverse Events) for pooling in the meta-analysis.

**Table 3. Quality assessment of the included studies.**

| | Newcastle-Ottawa scale | | | | |
|---|---|---|---|---|---|
| **Items** | **Score standard** | **Score** | **Study** | | |
| | | | **Huihui Yang 2021** | **Yizhao Xie 2021** | **Fei Chen 2021** |
| Selection | Representativeness of the exposed cohort | 1 | × | × | × |
| | Selection of the non-exposed cohort | 1 | √ | √ | √ |
| | Ascertainment of exposure | 1 | √ | √ | √ |
| | Demonstration that outcome of interest was not present at start of study | 1 | √ | √ | √ |
| Comparability | Comparability of cohorts on the basis of the design or analysis | 2 | √ | √ | √ |
| Exposure | Assessment of outcome | 1 | × | × | × |
| | Was follow-up long enough for outcomes to occur | 1 | √ | √ | √ |
| | Adequacy of follow-up of cohorts | 1 | √ | √ | × |
| | Total score | | 7 | 7 | 6 |

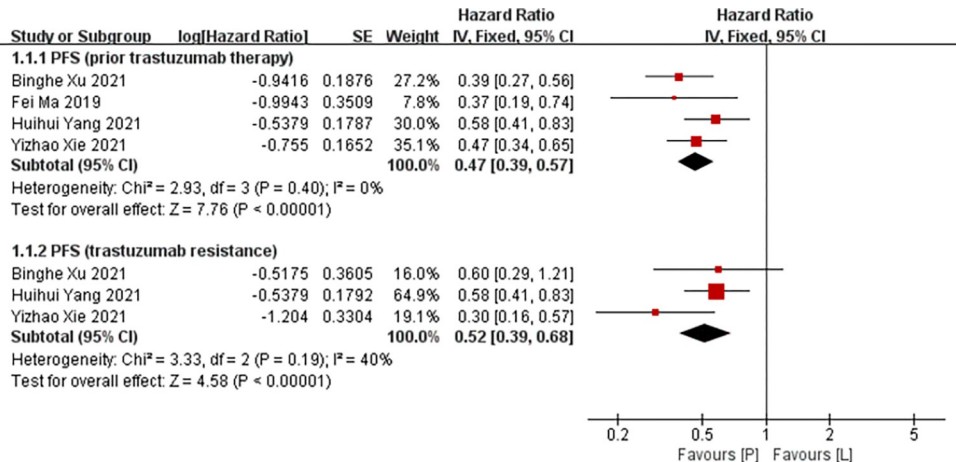

**Fig 4. Meta-analysis of PFS.**

*Diarrhoea*. Four studies [16,17,19,20] reported data on diarrhea for pooling in meta-analysis. Heterogeneity tests of p = 0.15 and $I^2$ = 44% were tested in diarrhea, showing a low heterogeneity. Therefore, fixed-effect models were used. The L arm showed significant improvements in diarrhea compared to the P arm (RR: 2.68, 95% CI: 1.85–3.90, *p*<0.001; Fig 6).

*Subgroup analysis*. The L arm showed significant improvements in diarrhea of RCTs compared to the P arm (RR: 3.58, 95% CI: 2.05–6.25, *p*<0.001). The P arm showed no statistical significance in diarrhea of non RCTs compared to the L arm (RR: 1.86, 95% CI: 0.75–4.62, *p* = 0.18). No subgroup differences were found (interaction test, *p* = 0.23).

*Hand-foot syndrome*. Three studies [16,17,19] reported data on hand-foot syndrome for pooling in meta-analysis. Heterogeneity tests of p = 0.07 and $I^2$ = 63% were tested in hand-foot syndrome, showing a high heterogeneity. Therefore, random-effect models were used. The P arm showed no statistical significance in hand-foot syndrome compared to the L arm (RR: 0.83, 95% CI: 0.39–1.75, *p* = 0.62; Fig 7).

*Vomiting*. Four studies [16,17,19,20] reported data on vomiting for pooling in meta-analysis. Heterogeneity tests of p = 0.33 and $I^2$ = 13% were tested in vomiting, showing a low heterogeneity. Therefore, fixed-effect models were used. The P arm showed no statistical significance in vomiting compared to the L arm (RR: 1.46, 95% CI: 0.62–3.47, *p* = 0.39; Fig 8).

*Subgroup analysis*. The P arm showed no statistical significance in vomiting of RCTs compared to the L arm (RR: 2.70, 95% CI: 0.87–8.32, *p* = 0.08). The P arm showed no statistical significance in vomiting of non RCTs compared to the L arm (RR: 0.49, 95% CI: 0.05–4.66, *p* = 0.54. No subgroup differences were found (interaction test, *p* = 0.18).

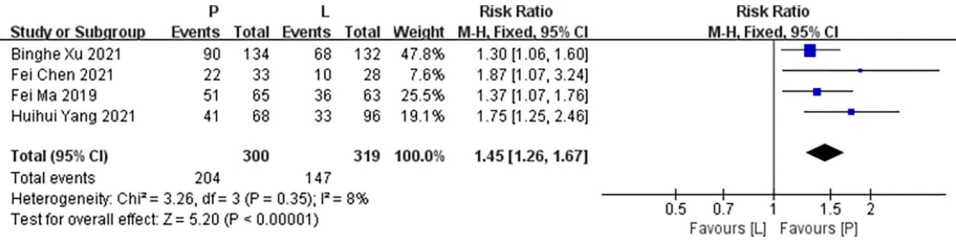

**Fig 5. Meta-analysis of ORR.**

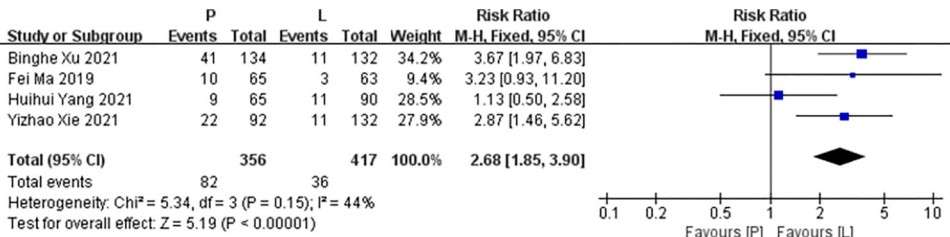

**Fig 6. Meta-analysis of grade ≥3 diarrhoea.**

*Nausea*. Four studies [16,17,19,20] reported data on nausea for pooling in meta-analysis. Heterogeneity tests of p = 0.46 and $I^2$ = 0% were tested in nausea, showing no heterogeneity. Therefore, fixed-effect models were used. The P arm showed no statistical significance in nausea compared to the L arm (RR: 1.20, 95% CI: 0.35–4.09, *p* = 0.77; Fig 9).

*Subgroup analysis*. The P arm showed no statistical significance in the nausea of RCTs compared to the L arm (RR: 3.84, 95% CI: 0.43–34.38, *p* = 0.07). The P arm showed no statistical significance in the nausea of non RCTs compared to the L arm (RR: 0.57, 95% CI: 0.08–4.25, *p* = 0.58). No subgroup differences were found (interaction test, *p* = 0.21).

*Neutropenia*. Four studies [16,17,19,20] reported data on neutropenia for pooling in meta-analysis. Heterogeneity tests of p = 0.77 and $I^2$ = 0% were tested in neutropenia, showing no heterogeneity. Therefore, fixed-effect models were used. The P arm showed no statistical significance in neutropenia compared to the L arm (RR: 1.78, 95% CI: 0.94–3.37, *p* = 0.08; Fig 10).

*Subgroup analysis*. The P arm showed no statistical significance in neutropenia of RCTs compared to the L arm (RR: 2.22, 95% CI: 0.93–5.30, *p* = 0.07). The P arm showed no statistical significance in vomiting of non RCTs compared to the L arm (RR: 1.29, 95% CI: 0.53–3.09, *p* = 0.57). No subgroup differences were found (interaction test, *p* = 0.39).

*Anemia*. Four studies [16,17,19,20] reported data on anemia for pooling in meta-analysis. Heterogeneity tests of p = 0.74 and $I^2$ = 0% were tested in anemia, showing no heterogeneity. Therefore, fixed-effect models were used. The P arm showed no statistical significance in anemia compared to the L arm (RR: 1.74, 95% CI: 0.47–6.49, *p* = 0.41; Fig 11).

*ALT increased*. Three studies [16,17,20] reported that data on ALT increased for pooling in meta-analysis. Heterogeneity tests of p = 0.36 and $I^2$ = 1% were tested in ALT, showing a low heterogeneity. Therefore, fixed-effect models were used. The P arm showed no statistical significance in leukopenia compared to the L arm (RR: 0.93, 95% CI: 0.25–3.39, *p* = 0.91; Fig 12).

*Leukopenia*. Three studies [16,17,20] reported data on leukopenia for pooling in meta-analysis. Heterogeneity tests of p = 0.04 and $I^2$ = 69% were tested in leukopenia, showing a high heterogeneity. Therefore, random-effect models were used. The P arm showed no statistical significance in leukopenia compared to the L arm (RR: 2.10, 95% CI: 0.43–10.38, *p* = 0.36; Fig 13).

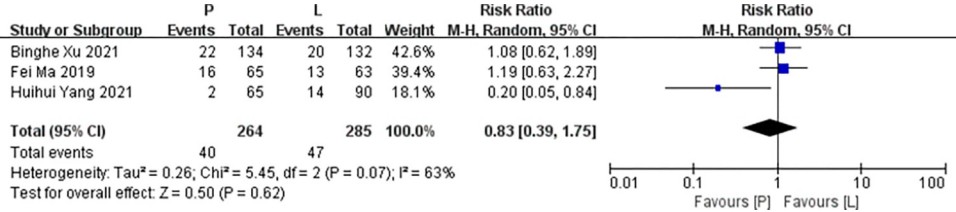

**Fig 7. Meta-analysis of grade ≥3 hand-foot syndrome.**

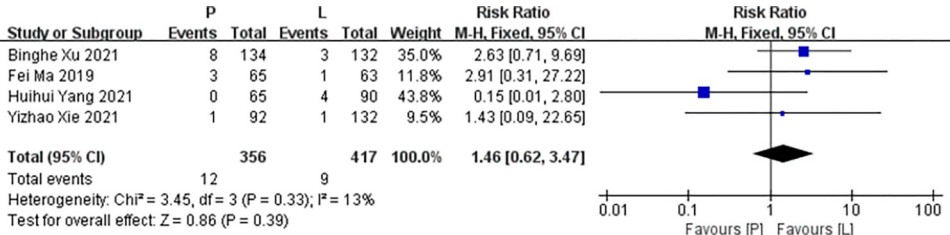

**Fig 8. Meta-analysis of grade ≥3 vomiting.**

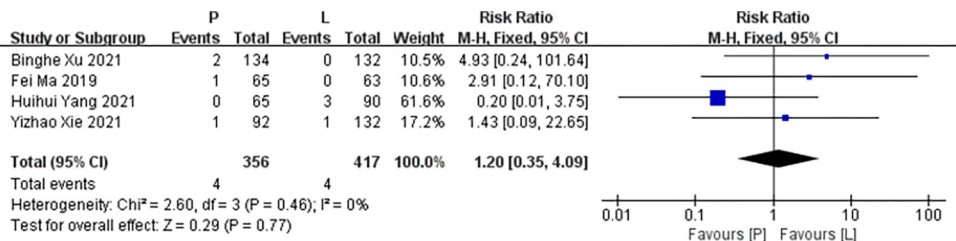

**Fig 9. Meta-analysis of grade ≥3 nausea.**

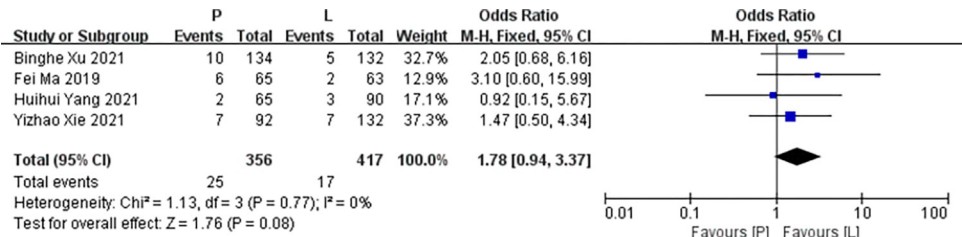

**Fig 10. Meta-analysis of grade ≥3 neutropenia.**

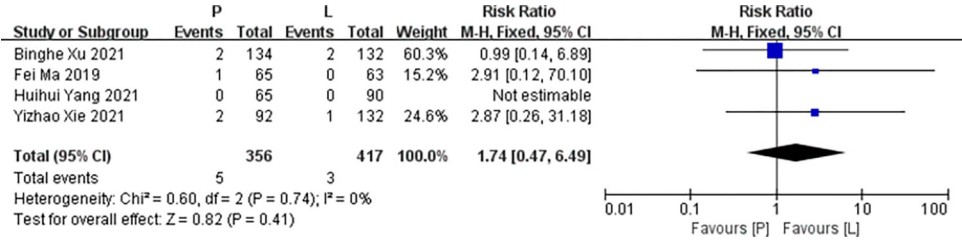

**Fig 11. Meta-analysis of grade ≥3 anemia.**

*Weight loss*. Three studies [16,17,20] reported data on weight loss for pooling in meta-analysis. Heterogeneity tests of p = 0.87 and $I^2$ = 0% were tested in weight loss, showing no heterogeneity. Therefore, fixed-effect models were used. The P arm showed no statistical significance in leukopenia compared to the L arm (RR: 3.56, 95% CI: 0.38–33.62, *p* = 0.27; Fig 14).

Findings from the meta-analysis are summarized in Table 4.

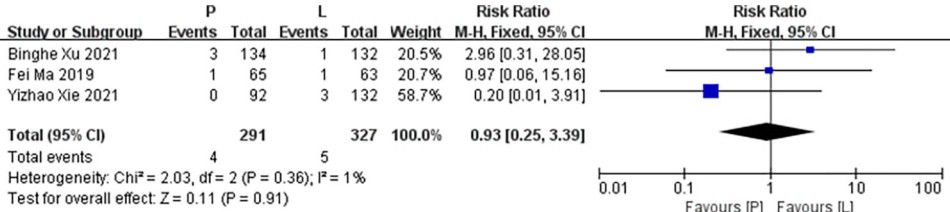

**Fig 12. Meta-analysis of grade ≥3 ALT increased.**

## Publication bias

We assessed publication bias for the primary outcomes. The Egger's test did not reveal any publication bias with regards to PFS (prior trastuzumab therapy) (t = -0.69, p = 0.559, p>0.05) (Fig 15) and PFS (trastuzumab resistance) (t = -0.59, p = 0.659, p>0.05) (Fig 16).

## Sensitivity analysis

The heterogeneity test for hand-foot syndrome (p = 0.07, $I^2$ = 63%) revealed a high heterogeneity. After excluding data from Huihui Yang's (2021) study with a low methodology method, there was no heterogeneity (p = 0.82, $I^2$ = 0). After deleting the heterogeneity source, the result of hand-foot syndrome using the fixed effects model revealed an insignificant difference from the previous result [RR = 1.13, 95% CI: 0.74–1.72, p = 0.58]. The heterogeneity test for leukopenia (p = 0.04, $I^2$ = 69%) revealed a high heterogeneity. After excluding data from Yizhao Xie's (2021) study with a low methodology method, there was no heterogeneity (p = 0.99, $I^2$ = 0). Therefore, this study is a source of heterogeneity. After deleting the heterogeneity source, the result of leukopenia using the fixed effects model revealed a significant difference from the previous result [RR = 4.90, 95% CI: 1.44–166, p = 0.01]. After removing all non RCTs, the result of leukopenia revealed a significant difference from the previous result [RR = 4.90, 95% CI: 1.44–166, p = 0.01]. After removing all RCTs, no outcomes showed a significant difference from the previous results. The details can be seen in Table 5.

## Evidence quality assessment

Three outcomes were assessed by GRADE. Risk bias: All outcomes were considered a severe risk due to the low methodology methods of the included studies. Inconsistency: Low heterogeneities were found in three outcomes. Thus, these outcomes were considered to have no risk of inconsistency. Indirectness: No outcome had a significant indirectness because all trials were direct comparisons. Imprecision: No outcome was considered a severe risk of imprecision due to insufficient sample size. Publication bias: No outcome exhibited a publication bias. Overall: All outcomes had moderate-quality evidence (Table 6).

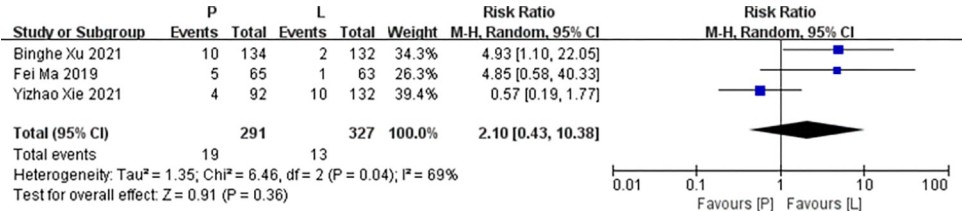

**Fig 13. Meta-analysis of grade ≥3 leukopenia.**

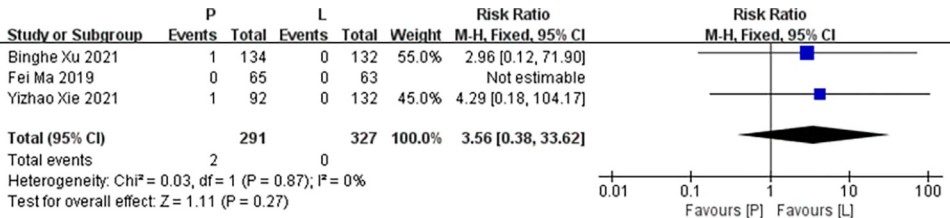

**Fig 14. Meta-analysis of grade ≥3 weight loss.**

## Discussion

It is critical to select subsequent treatments for patients after failure of trastuzumab therapy. Tyrosine kinase inhibitors (TKIs) may have some advantages in overcoming drug resistance because of their different mechanism of action with the monoclonal antibody, and guidelines of the Chinese Society of Clinical Oncology recommend that patients with HER2-positive metastatic breast cancer be treated with pyrotinib and lapatinib plus capecitabine following the failure of trastuzumab therapy [15,22]. In WJOG6110B/ELTOP, the lapatinib plus capecitabine arm showed more effective progression-free survival (PFS) (hazard ratio 0.81, 90% CI: 0.55–1.21) and overall survival (OS) (hazard ratio 0.58, 95% CI: 0.26–1.31) than trastuzumab plus capecitabine arm among metastatic breast cancer (MBC) patients those who had previous taxane treatment and progressed to trastuzumab-containing regimens [12]. In CEREBEL, the lapatinib plus capecitabine arm showed longer median PFS (6.6 months, 95% CI: 5.7–8.3) than the trastuzumab plus capecitabine arm (6.1 months, 95% CI: 5.7–8.0) among MBC patients previously treated with trastuzumab [13]. Currently, some studies proved that pyrotinib plus capecitabine showed more benefits than lapatinib plus capecitabine therapy but had more safety risks [16,17]. Fei Ma et al. reported that a higher overall response rate was found in the pyrotinib arm (78.5%, 95%CI: 68.5% to 88.5%) compared with the lapatinib arm (57.1%, 95% CI, 44.9% to 69.4%) [16]. In addition, some network meta-analyses [23–25] reported similar conclusions. However, the small number of relevant studies and sample size limited the reliability of the conclusions of these network meta-analyses. Apart from that, no meta-analysis

**Table 4. Summary of meta-analysis results.**

| Outcomes | | P vs. L | | | | |
|---|---|---|---|---|---|---|
| | | **HR/RR 95%CI** | **P-value** | **I²** | **Effect model** | **Superior arm** |
| Primary outcomes | PFS (prior trastuzumab therapy) | HR: 0.47, 95% CI: 0.39–0.57 | <0.001 | 0% | fixed | P |
| | PFS (trastuzumab resistance) | HR: 0.52, 95% CI: 0.39–0.68 | <0.001 | 40% | fixed | P |
| Secondary outcomes | ORR | RR: 1.45, 95% CI: 1.26–1.67 | <0.001 | 8% | fixed | P |
| Grade≥3 AEs | Diarrhoea | RR: 2.68, 95% CI: 1.85–3.90 | <0.001 | 44% | fixed | L |
| | Hand-foot syndrome | RR: 0.83, 95% CI: 0.39–1.75 | 0.62 | 63% | random | No statistical significance |
| | Vomiting | RR: 1.46, 95% CI: 0.62–3.47 | 0.39 | 13% | fixed | No statistical significance |
| | Nausea | RR: 1.20, 95% CI: 0.35–4.09 | 0.77 | 0% | fixed | No statistical significance |
| | Neutropenia | RR: 1.78, 95% CI: 0.94–3.37 | 0.08 | 0% | fixed | No statistical significance |
| | Anemia | RR: 1.74, 95% CI: 0.47–6.49 | 0.41 | 0% | fixed | No statistical significance |
| | ALT increased | RR: 0.93, 95% CI: 0.25–3.39 | 0.91 | 1% | fixed | No statistical significance |
| | Leukopenia | RR: 2.10, 95% CI: 0.43–10.38 | 0.36 | 69% | random | No statistical significance |
| | Weight loss | RR: 3.56, 95% CI: 0.38–33.62 | 0.27 | 0% | fixed | No statistical significance |

HR: Hazard ratio; RR: Risk ratio; 95%CI: 95% confidence interval; PFS: Progression free survival; ORR: Overall response rate; AEs: Adverse events.

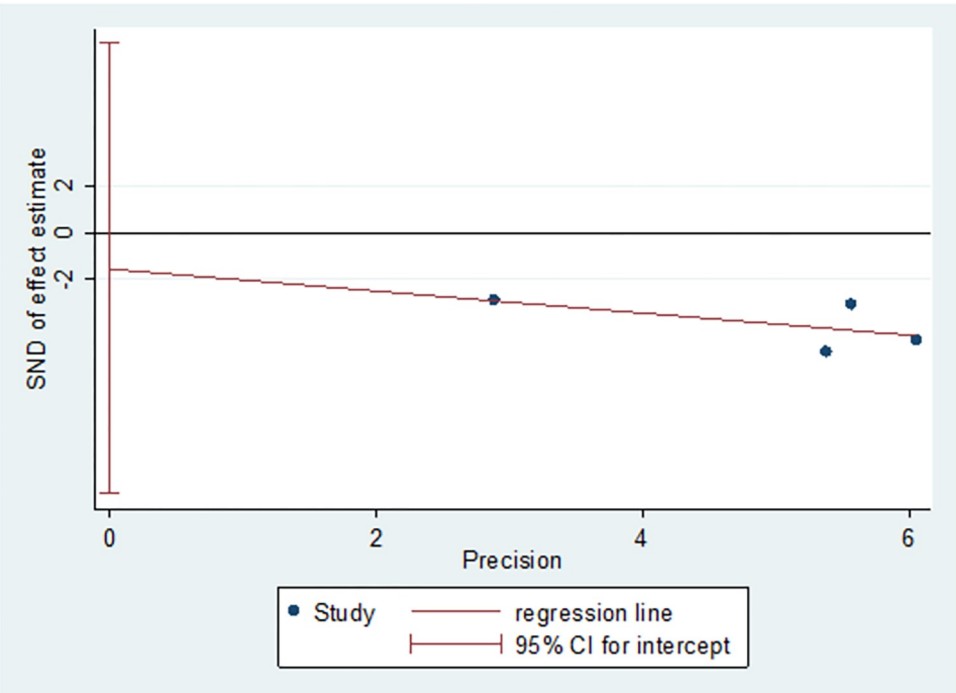

**Fig 15. Publication bias of PFS (prior trastuzumab therapy).**

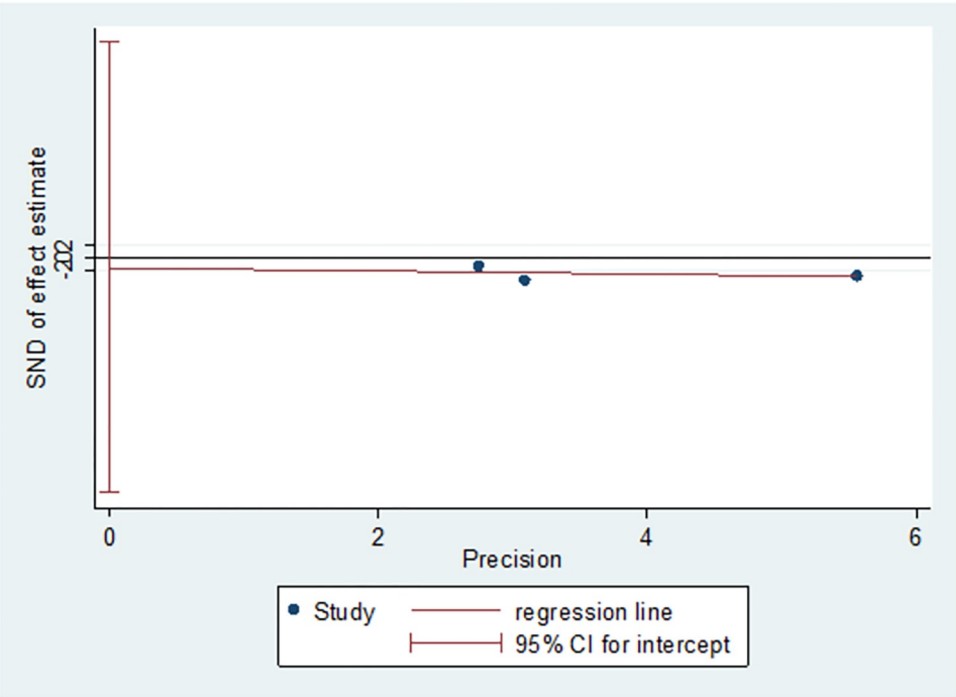

**Fig 16. Publication bias of PFS (trastuzumab resistance).**

**Table 5. Sensitivity analysis.**

| Trials | No. of patients | P | L | RR or HR (95% CI) | *P*-value | $I^2$ (%) |
|---|---|---|---|---|---|---|
| PFS (prior trastuzumab therapy) | | | | | | |
| Binghe Xu 2021 | 266 | 134 | 132 | 0.50 [0.40, 0.63] | <0.00001 | 15% |
| Fei Ma 2019 | 69 | 35 | 34 | 0.48 [0.39, 0.58] | <0.00001 | 18% |
| Huihui Yang 2021 | 155 | 65 | 90 | 0.43 [0.34, 0.54] | <0.00001 | 0% |
| Yizhao Xie 2021 | 224 | 92 | 132 | 0.47 [0.37, 0.59] | <0.00001 | 32% |
| Pooled estimate | 714 | 326 | 388 | 0.47 [0.39, 0.57] | <0.00001 | 0% |
| PFS (prior trastuzumab therapy) RCTs only | | | | | | |
| Binghe Xu 2021 | 266 | 134 | 132 | 0.37 [0.19, 0.74] | <0.00001 | / |
| Fei Ma 2019 | 69 | 35 | 34 | 0.39 [0.27, 0.56] | <0.00001 | / |
| Pooled estimate | 335 | 169 | 166 | 0.39 [0.28, 0.53] | <0.00001 | 0% |
| PFS (prior trastuzumab therapy) non RCTs only | | | | | | |
| Huihui Yang 2021 | 155 | 65 | 90 | 0.47 [0.34, 0.65] | <0.00001 | / |
| Yizhao Xie 2021 | 224 | 92 | 132 | 0.58 [0.41, 0.83] | 0.003 | / |
| Pooled estimate | 379 | 157 | 222 | 0.53 [0.39, 0.73] | <0.00001 | 0% |
| PFS (trastuzumab resistance) | | | | | | |
| Binghe Xu 2021 | 69 | 37 | 32 | 0.50 [0.37, 0.68] | <0.0001 | 68% |
| Huihui Yang 2021 | 155 | 65 | 90 | 0.41 [0.25, 0.66] | 0.0003 | 49% |
| Yizhao Xie 2021 | 59 | 30 | 29 | 0.59 [0.43, 0.80] | 0.0009 | 0% |
| Pooled estimate | 283 | 132 | 151 | 0.52 [0.39, 0.68] | <0.00001 | 40% |
| PFS (trastuzumab resistance) non RCTs only | | | | | | |
| Huihui Yang 2021 | 155 | 65 | 90 | 0.30 [0.16, 0.57] | 0.0003 | / |
| Yizhao Xie 2021 | 59 | 30 | 29 | 0.58 [0.41, 0.83] | 0.003 | / |
| Pooled estimate | 214 | 95 | 119 | 0.50 [0.37, 0.68] | <0.0001 | 68% |
| ORR | | | | | | |
| Binghe Xu 2021 | 266 | 134 | 132 | 1.58 [1.31, 1.92] | <0.00001 | 0% |
| Fei Chen 2021 | 61 | 33 | 28 | 1.42 [1.23, 1.64] | <0.00001 | 11% |
| Fei Ma 2019 | 128 | 65 | 63 | 1.48 [1.25, 1.74] | <0.00001 | 36% |
| Huihui Yang 2021 | 155 | 65 | 90 | 1.38 [1.18, 1.61] | <0.0001 | 0% |
| Pooled estimate | 610 | 297 | 313 | 1.45 [1.26, 1.67] | <0.00001 | 8% |
| ORR RCTs only | | | | | | |
| Binghe Xu 2021 | 266 | 134 | 132 | 1.37 [1.07, 1.76] | 0.01 | / |
| Fei Ma 2019 | 128 | 65 | 63 | 1.30 [1.06, 1.60] | 0.01 | / |
| Pooled estimate | 394 | 199 | 195 | 1.33 [1.13, 1.56] | 0.0004 | 0% |
| ORR non RCTs only | | | | | | |
| Fei Chen 2021 | 61 | 33 | 28 | 1.75 [1.25, 2.46] | 0.001 | / |
| Huihui Yang 2021 | 155 | 65 | 90 | 1.87 [1.07, 3.24] | 0.03 | / |
| Pooled estimate | 216 | 98 | 118 | 1.79 [1.34, 2.38] | <0.0001 | 0% |
| Diarrhoea | | | | | | |
| Binghe Xu 2021 | 266 | 134 | 132 | 2.17 [1.36, 3.47] | <0.00001 | 42% |
| Fei Ma 2019 | 128 | 65 | 63 | 2.63 [1.78, 3.88] | <0.00001 | 62% |
| Huihui Yang 2021 | 155 | 65 | 90 | 3.30 [2.15, 5.07] | <0.00001 | 0% |
| Yizhao Xie 2021 | 224 | 92 | 132 | 2.61 [1.67, 4.08] | <0.0001 | 62% |
| Pooled estimate | 773 | 356 | 417 | 2.68 [1.85, 3.90] | <0.00001 | 44% |
| Diarrhoea RCTs only | | | | | | |
| Binghe Xu 2021 | 266 | 134 | 132 | 3.23 [0.93, 11.20] | 0.06 | / |
| Fei Ma 2019 | 128 | 65 | 63 | 3.67 [1.97, 6.83] | <0.0001 | / |
| Pooled estimate | 394 | 199 | 195 | 3.58 [2.05, 6.24] | <0.00001 | 0% |

(*Continued*)

**Table 5.** (Continued)

| Trials | No. of patients | P | L | RR or HR (95% CI) | *P*-value | I$^2$ (%) |
|---|---|---|---|---|---|---|
| Diarrhoea non RCTs only | | | | | | |
| Huihui Yang 2021 | 155 | 65 | 90 | 2.87 [1.46, 5.62] | 0.002 | / |
| Yizhao Xie 2021 | 224 | 92 | 132 | 1.13 [0.50, 2.58] | 0.77 | / |
| Pooled estimate | 379 | 157 | 222 | 1.99 [1.20, 3.31] | 0.008 | 66% |
| Hand-foot syndrome | | | | | | |
| Binghe Xu 2021 | 266 | 134 | 132 | 0.54 [0.09, 3.37] | 0.51 | 81% |
| Fei Ma 2019 | 128 | 65 | 63 | 0.53 [0.10, 2.87] | 0.46 | 80% |
| Huihui Yang 2021 | 155 | 65 | 90 | 1.13 [0.74, 1.72] | 0.57 | 0% |
| Pooled estimate | 549 | 264 | 285 | 0.83 [0.39, 1.75] | 0.62 | 63% |
| Hand-foot syndrome RCTs only | | | | | | |
| Binghe Xu 2021 | 266 | 134 | 132 | 1.19 [0.63, 2.27] | 0.59 | / |
| Fei Ma 2019 | 128 | 65 | 63 | 1.08 [0.62, 1.89] | 0.77 | / |
| Pooled estimate | 394 | 199 | 195 | 1.13 [0.74, 1.72] | 0.57 | 0% |
| Vomiting | | | | | | |
| Binghe Xu 2021 | 266 | 134 | 132 | 0.84 [0.24, 2.90] | 0.78 | 25% |
| Fei Ma 2019 | 128 | 65 | 63 | 1.27 [0.49, 3.29] | 0.62 | 38% |
| Huihui Yang 2021 | 155 | 65 | 90 | 2.48 [0.88, 7.02] | 0.09 | 0% |
| Yizhao Xie 2021 | 224 | 92 | 132 | 1.47 [0.59, 3.65] | <0.41 | 42% |
| Pooled estimate | 773 | 356 | 417 | 1.46 [0.62, 3.47] | 0.39 | 13% |
| Vomiting RCTs only | | | | | | |
| Binghe Xu 2021 | 266 | 134 | 132 | 2.91 [0.31, 27.22] | 0.35 | / |
| Fei Ma 2019 | 128 | 65 | 63 | 2.63 [0.71, 9.69] | 0.15 | / |
| Pooled estimate | 394 | 199 | 195 | 2.70 [0.87, 8.33] | 0.08 | 0% |
| Vomiting non RCTs only | | | | | | |
| Huihui Yang 2021 | 155 | 65 | 90 | 1.43 [0.09, 22.65] | 0.80 | / |
| Yizhao Xie 2021 | 224 | 92 | 132 | 0.15 [0.01, 2.80] | 0.21 | / |
| Pooled estimate | 379 | 157 | 222 | 0.38 [0.06, 2.28] | 0.29 | 21% |
| Leukopenia | | | | | | |
| Binghe Xu 2021 | 266 | 134 | 132 | 1.38 [0.17, 10.99] | 0.76 | 68% |
| Fei Ma 2019 | 128 | 65 | 63 | 1.59 [0.19, 13.24] | 0.67 | 81% |
| Yizhao Xie 2021 | 224 | 92 | 132 | 4.90 [1.44, 16.66] | 0.01 | 0% |
| Pooled estimate | 618 | 291 | 327 | 2.10 [0.43, 10.38] | 0.36 | 69% |
| Leukopenia RCTs only | | | | | | |
| Binghe Xu 2021 | 266 | 134 | 132 | 4.85 [0.58, 40.33] | 0.14 | / |
| Fei Ma 2019 | 128 | 65 | 63 | 4.93 [1.10, 22.05] | 0.04 | / |
| Pooled estimate | 394 | 199 | 195 | 4.90 [1.44, 16.66] | 0.01 | 0% |
| Neutropenia | | | | | | |
| Binghe Xu 2021 | 266 | 134 | 132 | 1.64 [0.75, 3.62] | 0.22 | 0% |
| Fei Ma 2019 | 128 | 65 | 63 | 1.58 [0.78, 3.19] | 0.20 | 0% |
| Huihui Yang 2021 | 155 | 65 | 90 | 1.95 [0.98, 3.90] | 0.06 | 0% |
| Yizhao Xie 2021 | 224 | 92 | 132 | 1.96 [0.88, 4.35] | 0.10 | 0% |
| Pooled estimate | 773 | 356 | 417 | 1.78 [0.94, 3.37] | 0.08 | 0% |
| Neutropenia RCTs only | | | | | | |
| Binghe Xu 2021 | 266 | 134 | 132 | 3.10 [0.60, 15.99] | 0.18 | / |
| Fei Ma 2019 | 128 | 65 | 63 | 2.05 [0.68, 6.16] | 0.20 | / |
| Pooled estimate | 394 | 199 | 195 | 2.35 [0.94, 5.84] | 0.07 | 0% |
| Neutropenia non RCTs only | | | | | | |

(*Continued*)

**Table 5.** (Continued)

| Trials | No. of patients | P | L | RR or HR (95% CI) | P-value | I$^2$ (%) |
|---|---|---|---|---|---|---|
| Huihui Yang 2021 | 155 | 65 | 90 | 1.47 [0.50, 4.34] | 0.49 | / |
| Yizhao Xie 2021 | 224 | 92 | 132 | 0.92 [0.15, 5.67] | 0.93 | / |
| Pooled estimate | 279 | 157 | 222 | 1.30 [0.51, 3.28] | 0.58 | 0% |
| Anemia | | | | | | |
| Binghe Xu 2021 | 266 | 134 | 132 | 2.88 [0.43, 19.50] | 0.28 | 0% |
| Fei Ma 2019 | 128 | 65 | 63 | 1.53 [0.36, 6.57] | 0.57 | 0% |
| Huihui Yang 2021 | 155 | 65 | 90 | 1.74 [0.47, 6.49] | 0.41 | 0% |
| Yizhao Xie 2021 | 224 | 92 | 132 | 1.37 [0.27, 6.89] | 0.70 | 0% |
| Pooled estimate | 713 | 356 | 417 | 1.74 [0.47, 6.49] | 0.41 | 0% |
| Anemia RCTs only | | | | | | |
| Binghe Xu 2021 | 266 | 134 | 132 | 2.91 [0.12, 70.10] | 0.51 | / |
| Fei Ma 2019 | 128 | 65 | 63 | 0.99 [0.14, 6.89] | 0.99 | / |
| Pooled estimate | 394 | 199 | 195 | 1.37 [0.27, 6.89] | 0.70 | 0% |
| Anemia non RCTs only | | | | | | |
| Huihui Yang 2021 | 155 | 65 | 90 | 2.87 [0.26, 31.18] | 0.39 | / |
| Yizhao Xie 2021 | 224 | 92 | 132 | / | / | / |
| Pooled estimate | 379 | 157 | 222 | 2.87 [0.26, 31.18] | 0.39 | / |
| ALT increased | | | | | | |
| Binghe Xu 2021 | 266 | 134 | 132 | 0.40 [0.06, 2.72] | 0.35 | 0% |
| Fei Ma 2019 | 128 | 65 | 63 | 0.92 [0.21, 3.99] | 0.91 | 51% |
| Yizhao Xie 2021 | 224 | 92 | 132 | 1.96 [0.36, 10.56] | 0.43 | 0% |
| Pooled estimate | 618 | 291 | 327 | 0.93 [0.25, 3.39] | 0.91 | 1% |
| ALT increased RCTs only | | | | | | |
| Binghe Xu 2021 | 266 | 134 | 132 | 0.97 [0.06, 15.16] | 0.98 | / |
| Fei Ma 2019 | 128 | 65 | 63 | 2.96 [0.31, 28.05] | 0.35 | / |
| Pooled estimate | 394 | 199 | 195 | 1.96 [0.36, 10.56] | 0.43 | 0% |
| Nausea | | | | | | |
| Binghe Xu 2021 | 266 | 134 | 132 | 0.76 [0.17, 3.29] | 0.71 | 0% |
| Fei Ma 2019 | 128 | 65 | 63 | 0.99 [0.25, 3.88] | 0.99 | 13% |
| Huihui Yang 2021 | 155 | 65 | 90 | 2.80 [0.52, 14.96] | 0.23 | 0% |
| Yizhao Xie 2021 | 224 | 92 | 132 | 1.15 [0.29, 4.53] | 0.84 | 23% |
| Pooled estimate | 773 | 356 | 417 | 1.20 [0.35, 4.09] | 0.77 | 0% |
| Nausea RCTs only | | | | | | |
| Binghe Xu 2021 | 266 | 134 | 132 | 2.91 [0.12, 70.10] | 0.51 | / |
| Fei Ma 2019 | 128 | 65 | 63 | 4.93 [0.24, 101.64] | 0.30 | / |
| Pooled estimate | 394 | 199 | 195 | 3.91 [0.44, 34.66] | 0.22 | 0% |
| Nausea non RCTs only | | | | | | |
| Huihui Yang 2021 | 155 | 65 | 90 | 1.43 [0.09, 22.65] | 0.80 | / |
| Yizhao Xie 2021 | 224 | 92 | 132 | 0.20 [0.01, 3.75] | 0.28 | / |
| Pooled estimate | 379 | 157 | 222 | 0.47 [0.07, 2.91] | 0.42 | 0% |
| Weight loss | | | | | | |
| Binghe Xu 2021 | 266 | 134 | 132 | 4.29 [0.18, 104.17] | 0.37 | / |
| Fei Ma 2019 | 128 | 65 | 63 | 3.56 [0.38, 33.62] | 0.27 | 0% |
| Yizhao Xie 2021 | 224 | 92 | 132 | 2.96 [0.12, 71.90] | 0.51 | / |
| Pooled estimate | 618 | 291 | 327 | 3.56 [0.38, 33.62] | 0.27 | 0% |
| Weight loss RCTs only | | | | | | |
| Binghe Xu 2021 | 266 | 134 | 132 | / | / | / |

(*Continued*)

**Table 5.** (Continued)

| Trials | No. of patients | P | L | RR or HR (95% CI) | *P*-value | $I^2$ (%) |
|---|---|---|---|---|---|---|
| Fei Ma 2019 | 128 | 65 | 63 | 2.96 [0.12, 71.90] | 0.51 | / |
| Pooled estimate | 394 | 199 | 195 | 2.96 [0.12, 71.90] | 0.51 | / |

directly compares pyrotinib therapy with lapatinib therapy. Thus, this study investigates whether pyrotinib is superior to lapatinib in efficacy and safety among HER2-positive MBC patients.

This meta-analysis shows that pyrotinib therapy is superior to lapatinib therapy among HER2+ metastatic breast cancer, with a significant improvement in PFS and ORR, but has more grade ≥3 diarrhea risks. Three network meta-analyses [23–25] found similar results. Pyrotinib showed significant improvement in PFS compared to lapatinib. Xinghui Li et al. and Hao Liao et al. [24,25] reported that pyrotinib might have more grade ≥3 adverse events risks. Hao Liao et al. [24] reported that pyrotinib had a higher probability of better ORR than lapatinib therapy.

PFS is a significant clinical outcome for clinicians and patients. We think it is even more critical than OS. When a patient progresses with breast cancer using a drug, other drugs will be used for the next stage of therapy with a high possibility. Thus, treatment with a longer PFS is extraordinarily significant. Fei Ma et al. reported that the longer median PFS was found in the pyrotinib arm (18.1 months, 95%CI, 13.9 months to not reached) compared with the lapatinib arm (7.0 months, 95% CI: 5.6–9.8 months) when treated prior trastuzumab therapy [16]. In PHOEBE, patients with pyrotinib showed significantly longer PFS than patients with lapatinib (hazard ratio 0.39, 95% CI: 0.27–0.56), and the median PFS of the pyrotinib arm was 12.5 months (95% CI: 9.7–not reached), and the median PFS of the lapatinib arm was 6.8 months (95% CI: 5.4–8.1) [17]. In this study, we pooled the effects of patients with prior trastuzumab therapy and trastuzumab resistance and found that the pyrotinib arm had a longer PFS than the lapatinib arm. Pyrotinib is a novel irreversible tyrosine kinase inhibitor with a different mechanism involved in pyrotinib anti-HER2 activity than trastuzumab. Pyrotinib exerts its anti-HER2 activity by directly targeting the intracellular tyrosine kinase region and blocking the downstream HER family homo/heterodimers pathways to cancer [14]. In these cases, pyrotinib might still be effective for patients with HER2-positive MBC who have progressed on trastuzumab [26]. The evidence quality for PFS is moderate, based on GRADE. Therefore, we suppose pyrotinib is a better treatment recommended for increasing the PFS of patients treated with trastuzumab previously and patients with trastuzumab resistance compared to lapatinib therapy. In addition, we find there is no difference between the meta-analysis results of PFS (prior trastuzumab therapy) and PFS (trastuzumab resistance). The patients with prior

**Table 6. GRADE evidence profile of outcomes.**

| Outcome | Number of studies | Assessment of evidence quality | | | | | Number of participants | Effect (95%CI) | Evidence quality |
|---|---|---|---|---|---|---|---|---|---|
| | | risk bias | inconsistency | indirectness | imprecision | publication bias | | | |
| PFS (prior trastuzumab therapy) | 4 | serious | no | no | no | undetected | 723 | HR = 0.47 [0.39, 0.57] | moderate |
| PFS (trastuzumab resistance) | 3 | serious | no | no | no | undetected | 292 | HR = 0.52 [0.39, 0.68] | moderate |
| ORR | 4 | serious | no | no | no | undetected | 619 | HR = 1.45 [1.26, 1.67] | moderate |

trastuzumab therapy include trastuzumab resistance and no trastuzumab resistance. We infer that the condition of trastuzumab resistance is not a factor that influences the efficacy of pyrotinib, and these two kinds of patients can both benefit from pyrotinib. Perhaps, more trials can be conducted to compare pyrotinib with trastuzumab among patients without prior trastuzumab therapy.

In China, some short-term studies have been performed to evaluate the efficacy of pyrotinib and trastuzumab and proved that pyrotinib had better short-term efficacy [27–29]. In Binghe Xu 2021, the median OS of the P arm was 26.8 months (95% CI: 26.2–not reached), and the L arm was not reached (21.8–not reached). While, in Fei Ma 2019, the median OS of the P arm was not reached (95% CI 26.3–not reached) and of the L arm was 29.9 months (23.7–not reached). In addition, only Fei Ma 2019 reported the HR of OS (HR: 0.69, 95% CI: 0.40–1.19). More studies are needed to assess the two kinds of treatment OS. In short-term efficacy, pyrotinib plus chemotherapy has a higher ORR than lapatinib plus chemotherapy. It means pyrotinib combined with chemotherapy has better efficacy for at least three months than lapatinib combined with chemotherapy. The evidence quality for ORR is moderate, based on GRADE. We think a good short-term efficacy could make patients more positive and confident about the treatment, and pyrotinib performs better in this aspect. Above all, whether long-term or short-term efficacy outcomes, pyrotinib is a superior option to lapatinib.

In safety outcomes, the pyrotinib arm showed a higher incidence of grade $\geq$3 diarrhea than the lapatinib arm. Although pyrotinib combined with chemotherapy causes more grade $\geq$3 diarrhea, the incidence of diarrhea decreased throughout the treatment process, and it is vital to educate patients, adjust their diets, and promptly treat symptoms with loperamide and Montmorillonite Powder when necessary [30]. In sensitivity analysis, meta-analysis result of leukopenia seems unstable. When a study with a low methodology is removed, the RR of leukopenia gets significantly higher and has a statistical significance. This may mention that pyrotinib leads to a higher grade$\geq$3 leukopenia incidence than lapatinib. However, we do not need to be so panic about adverse events. In phase I/II studies, pyrotinib has proved to be clinically effective and tolerable [31,32].

Various anti-HER2 drugs have been used in clinical trials in recent years. Therefore, it is important to choose a safe and effective therapy. To our knowledge, this study is the first meta-analysis comparing pyrotinib combined with chemotherapy with lapatinib directly. This study includes more relevant studies and enlarges the sample size to prove the meta-analysis results more reliable and scientific. However, the limitations exist. First, the relevant studies are still inadequate, especially RCTs. Second, almost all trials were conducted at hospitals in China, and most participants were Chinese. The lack of data from other countries and races may make the meta-analysis incomplete. Finally, publication bias and information bias may exist because the included studies are published in Chines and English only.

## Conclusion

The efficacy of pyrotinib combined with chemotherapy is superior to lapatinib combined with chemotherapy but has more safety risks. In the future, relevant, well-designed, and long-term large sample RCTs are needed, and more trials are necessary for other countries and races.

## Author Contributions

**Formal analysis:** Yi Cai.

**Investigation:** Xumei Liu.

**Methodology:** Xumei Liu.

**Software:** Xumei Liu.

**Writing – original draft:** Ye Yuan.

**Writing – review & editing:** Ye Yuan, Wenyuan Li.

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
