## [Decision Letter · Decision Letter 0]

21 Nov 2022

PONE-D-22-24109Pyrotinib versus lapatinib therapy for HER2 positive metastatic breast cancer patients: A systematic review and meta-analysisPLOS ONE

Dear Dr. Yuan,

Thank you for submitting your manuscript to PLOS ONE. After careful consideration, we feel that it has merit but does not fully meet PLOS ONE’s publication criteria as it currently stands. Therefore, we invite you to submit a revised version of the manuscript that addresses the points raised during the review process.

We look forward to receiving your revised manuscript.

Kind regards,

Nabil Elhadi Elsayed Ali Omar, PharmD.,BCOP.,PhD(C)

Academic Editor

PLOS ONE

Journal Requirements:

Reviewers' comments:

Reviewer's Responses to Questions

**Comments to the Author**

1. Is the manuscript technically sound, and do the data support the conclusions?

Reviewer #1: Yes

Reviewer #2: Yes

2. Has the statistical analysis been performed appropriately and rigorously? 

Reviewer #1: Yes

Reviewer #2: Yes

3. Have the authors made all data underlying the findings in their manuscript fully available?

Reviewer #1: No

Reviewer #2: Yes

4. Is the manuscript presented in an intelligible fashion and written in standard English?

Reviewer #1: Yes

Reviewer #2: Yes

5. Review Comments to the Author

Reviewer #1: dear authors, this is an interesting work on new HER2-directed TKI relevant for the use in China, post-trastuzumab. While data on post-pertuzumab or other HER2 drugs approved in US are not available, the trials reported well fit with the local situation. Comparison with lapatinib is appropriate.

The authors should emphasize that this is a combined analysis of randomized and non-randomized trials. They should report in the sensitivity analysis, the finding in randomized versus non randomized trials, to ensure consistency.

Reviewer #2: The research idea is novel, and it is important to the field of oncology. I appreciate your efforts to do this SRMA. The language and structure of the manuscript are appropriate. I have some comments that I hope may improve your paper.

Language: There are some writings that need improvement. For instance, the abbreviation AEs is used without writing what it is referring to.

Title: The title is clear and included all important elements, I would suggest adding that it is “after first-line treatment failure”

Introduction: There are many statistical results mentioned in the introduction, I prefer to use them in the discussion and show fewer numbers in the introduction.

Rationale: Valid and well described including existing knowledge and evidence

Objective: I would prefer to see a clear objective of why you would like to do SRMA. Since it is reported as different objective, efficacy and safety should be considered different objectives and that should be stated clearly in the introduction

Methods:

I appreciate that you used Chinese search terms and did not limit the search to English only.

Inclusion criteria: did you include only articles that studied the two medications as second-line after treatment failure?

Statistical analysis and models used in the SRMA were well established and explained in the context.

Please specify the outcomes to be assessed in the Methods section. In results, I found that you have defined primary and secondary outcomes, but that was not mentioned in the methods. Also, you did not mention that you will assess the adverse events which surprisingly appeared in the results too. I recommend that these outcomes and mentioned first in the methodology.

Results:

I would prefer to include the data in tables to make it easier for the reader. A table for the primary outcomes, a table for the secondary outcomes, and a table for the adverse events.

Discussion: It is relevant and well written, however, there is a great focus on the safety outcomes, I suggest shortening this paragraph and working more on the primary outcomes. In addition, I recommend separating the conclusion for the discussion, adding a conclusion section where you summarise the findings and suggest future studies would be helpful for the reader.

6. PLOS authors have the option to publish the peer review history of their article (what does this mean?). If published, this will include your full peer review and any attached files.

Reviewer #1: No

Reviewer #2: **Yes: **

---

## [Author Response · Author response to Decision Letter 0]

26 Nov 2022

1.Please ensure that your manuscript meets PLOS ONE's style requirements, including those for file naming.

Answer: I have changed style of my manuscript totally. The figures were re-uploaded with new file names.

2.Please upload a copy of Figures 11, 12 and 19 to which you refer in your text on page 7 and 8. If the figure is no longer to be included as part of the submission please remove all reference to it within the text.

Answer: I am sorry for my careless. I have resubmitted the Figs 11 and 12 (see page 12, lines 6 and 14) in the manuscript. All reference to figure 19 within the text was removed.

3.Please ensure that you refer to Figures 50, 61 and 72 in your text as, if accepted, production will need this reference to link the reader to the figure.

Answer: I am so sorry. It is a mistake. The wrong numbers were named to these figures. I have corrected them in revised manuscript. 

4.We note you have included a table to which you do not refer in the text of your manuscript. Please ensure that you refer to Table 4 in your text; if accepted, production will need this reference to link the reader to the Table.

Answer: Table 4 was Sensitivity analysis. It is Table 5 now (see page 15, line 13), because a new table was added. 

5.Please review your reference list to ensure that it is complete and correct. If you have cited papers that have been retracted, please include the rationale for doing so in the manuscript text, or remove these references and replace them with relevant current references. Any changes to the reference list should be mentioned in the rebuttal letter that accompanies your revised manuscript. If you need to cite a retracted article, indicate the article’s retracted status in the References list and also include a citation and full reference for the retraction notice.

Answer: Thanks for your comment. I reviewed my reference list. I found reference 14 was same with reference 27, and reference 20 was same with reference 23. Thus, I removed the references 23 and 27. I removed some contents according to the comments of reviewer 2. Thus, the relevant references (references 30, 34 and 35) were removed either. The rest of references can be found in PubMed and Chinese electronic databases, like CNKI and Wan Fang.

1.Is the manuscript technically sound, and do the data support the conclusions?

Reviewer #1: Yes

Reviewer #2: Yes

2. Has the statistical analysis been performed appropriately and rigorously?

Reviewer #1: Yes

Reviewer #2: Yes

3. Have the authors made all data underlying the findings in their manuscript fully available?

Reviewer #1: No

Reviewer #2: Yes

Answer: Thanks for your comments. All data can be found in public repository without restriction in this manuscript. I will provide all PDF of included studies, if there is any need.

4. Is the manuscript presented in an intelligible fashion and written in standard English?

Reviewer #1: Yes

Reviewer #2: Yes

Reviewer #1: dear authors, this is an interesting work on new HER2-directed TKI relevant for the use in China, post-trastuzumab. While data on post-pertuzumab or other HER2 drugs approved in US are not available, the trials reported well fit with the local situation. Comparison with lapatinib is appropriate.

The authors should emphasize that this is a combined analysis of randomized and non-randomized trials. They should report in the sensitivity analysis, the finding in randomized versus non randomized trials, to ensure consistency.

Answer: It is a very important comment for us! I have re-performed my sensitity analysis after removing RCTs and non-RCTs (See page 15, lines 6-11). In addition, the table of sensitivity analysis was changed too (Table 5).

Reviewer #2: The research idea is novel, and it is important to the field of oncology. I appreciate your efforts to do this SRMA. The language and structure of the manuscript are appropriate. I have some comments that I hope may improve your paper.

Language: There are some writings that need improvement. For instance, the abbreviation AEs is used without writing what it is referring to.

Answer: Thanks for your comment. I have corrected these mistakes (see page 3, line 17; page 9, line 1; page 10, line 7; page 11, lines 3, 16 and 29).

Title: The title is clear and included all important elements, I would suggest adding that it is “after first-line treatment failure”

Answer: Your suggestion is very important. I have added “after first-line treatment failure” in my title. I removed “A systematic review and meta-analysis”, because this title seems too long.

Introduction: There are many statistical results mentioned in the introduction, I prefer to use them in the discussion and show fewer numbers in the introduction.

Answer: I have deleted the statistical results mentioned in the introduction (see page 2, lines 18-23 and 29-32) and added them in my discussion (see page 20, lines 9-15, 17-19 and 33-38).

Rationale: Valid and well described including existing knowledge and evidence

Answer: Thank you for your comment.

Objective: I would prefer to see a clear objective of why you would like to do SRMA. Since it is reported as different objective, efficacy and safety should be considered different objectives and that should be stated clearly in the introduction

Answer: Thanks for your comment. I have reported my objective again in the introduction (see page 2, lines 36-38). 

Methods:

I appreciate that you used Chinese search terms and did not limit the search to English only.

Inclusion criteria: did you include only articles that studied the two medications as second-line after treatment failure?

Answer: Yes, I did. I added this in my inclusion criteria (see page 3, lines 18-19).

Statistical analysis and models used in the SRMA were well established and explained in the context.

Please specify the outcomes to be assessed in the Methods section. In results, I found that you have defined primary and secondary outcomes, but that was not mentioned in the methods. Also, you did not mention that you will assess the adverse events which surprisingly appeared in the results too. I recommend that these outcomes and mentioned first in the methodology.

Answer: Thank you, this comment is very helpful. I have added these outcomes in the methodology (see page 3, lines 16-17).

Results:

I would prefer to include the data in tables to make it easier for the reader. A table for the primary outcomes, a table for the secondary outcomes, and a table for the adverse events.

Answer: Thanks for your comment. I have added a new table where findings from the meta-analysis are summarized (see page 13, line 9).

Discussion: It is relevant and well written, however, there is a great focus on the safety outcomes, I suggest shortening this paragraph and working more on the primary outcomes. In addition, I recommend separating the conclusion for the discussion, adding a conclusion section where you summarise the findings and suggest future studies would be helpful for the reader.

Answer: Thanks for your comment. I have shorten the content about safety (see page 21, lines 24-32) and worked more on the primary outcomes (see page 20, lines 33-38). In addition, I have separated the conclusion (see page 22, lines 1-3).

---

## [Editor Report · Decision Letter 1]

14 Dec 2022

Pyrotinib versus lapatinib therapy for HER2 positive metastatic breast cancer patients after first-line treatment failure

PONE-D-22-24109R1

Dear Dr. Ye Yuan

We’re pleased to inform you that your manuscript has been judged scientifically suitable for publication and will be formally accepted for publication once it meets all outstanding technical requirements.

Kind regards,

Nabil Elhadi Elsayed Ali Omar, PharmD.,BCOP.,PhD(C)

Academic Editor

PLOS ONE

Additional Staff Editor Comments :  "Please confirm that you have identified the study as a meta-analysis or systematic review in the title."
---

## [Editor Report · Acceptance letter]

20 Dec 2022

PONE-D-22-24109R1 

Pyrotinib versus lapatinib therapy for HER2 positive metastatic breast cancer patients after first-line treatment failure: a meta-analysis and systematic review 

Dear Dr. Yuan:

I'm pleased to inform you that your manuscript has been deemed suitable for publication in PLOS ONE. Congratulations! Your manuscript is now with our production department. 

Kind regards, 

on behalf of

Dr. Nabil Elhadi Elsayed Ali Omar 

Academic Editor

PLOS ONE